# Effect of Ohmic Heating on the Extraction Yield, Polyphenol Content and Antioxidant Activity of Olive Mill Leaves

Fereshteh Safarzadeh Markhali [1,2,*], José A. Teixeira [1,2] and Cristina M. R. Rocha [1,2]

1   CEB–Centre of Biological Engineering, Campus of Gualtar, University of Minho, 4710-057 Braga, Portugal; jateixeira@deb.uminho.pt (J.A.T.); cmrocha@ceb.uminho.pt (C.M.R.R.)
2   LABBELS–Associate Laboratory, 4710-057 Braga, Portugal
*   Correspondence: id7987@alunos.uminho.pt or fsafar10@caledonian.ac.uk

**Abstract:** This study examined the influence of ohmic heating (*OH*), compared to the conventional heating (*Conven*) and *Control* (solvent) methods, on the extraction of olive mill leaves. The main extraction parameters were: (i) solvent ratio (aqueous ethanol; 40%, 60%, and 80%, *v/v*), and (ii) extraction temperature; 45 °C, 55 °C, and 75 °C (for *OH* and *Conven*), and room temperature (for *Control*). The selected response variables were extraction yield (%), total phenolic content (TPC), and antioxidant activity (ABTS and DPPH). The ohmic system, compared to *Conven* and *Control*, exhibited the greatest effects ($p < 0.001$) on increasing (i) extraction yield (34.53%) at 75 °C with 80% ethanol, (ii) TPC at 55 °C (42.53, 34.35, 31.63 mg GAE/g extract, with 60%, 40%, and 80% ethanol, respectively), and (iii) antioxidant potency at 75 °C detected by DPPH and ABTS, in the range of 1.21–1.04 mM TE/g, and 0.62–0.48 mM TE/g extract, respectively. Further, there were relatively similar trends in TPC and antioxidant activity (both methods), regardless of solvent ratios, $p < 0.001$. These findings demonstrate the potential of ohmic heating, as a green processing tool, for efficient extraction (15 min) of olive leaves. To date, no literature has described ohmic application for olive leave extraction.

**Keywords:** ohmic heating; olive leaves; residual biomass; polyphenols; antioxidant activity; green solvent; sustainable extraction

## 1. Introduction

Olive leaves constitute a large proportion of the residual biomass generated from agro-industrial activities of olive crops and, while being abundant in high-added-value ingredients, currently have found low-value applications in folk/traditional medicine [1] and animal feed [2]. Polyphenols are among the key antioxidants in olive leaves, markedly prized for their bio-functional potentials, and this fact has prompted continuous research studies to devise optimum processing design to maximize their recovery at a low-cost/durable system. The conventional methods, such as maceration, Soxhlet, and percolation, although being rather simple, are inherently (i) inefficient in the processing/extraction system, (ii) solvent/energy intensive, and (iii) time consuming. To address the challenges associated with the existing technology, the emphasis is often placed on the optimization of emerging/green technologies, including microwave-assisted extraction (MAE), ultrasonic-assisted extraction (UAE), supercritical fluid extraction (SFE), pulsed electric field (PEF), and high-voltage electric discharges (HVED).

In the research of Da Rosa et al. [3], the extraction efficiency between MAE, UAE, and maceration (conventional) was compared and the results confirmed that the use of MAE, particularly over the conventional method, was significantly productive. Le Floch et al. [4] detected a higher concentration of total phenolics using SFE compared to that obtained by sonication-assisted liquid solvent extraction (with the exception of methanol solvent). The increased phenolic recovery from olive leaves by means of HVED was well justified in the study of Žuntar et al. [5], highlighting that the use of this method exerts effects on

the increased cell rupture and mass transfer within a short extraction time through using high-voltage pulsed electric field. Pappas et al. [6] found increased total phenolics through the extraction of olive leaves by PEF (using 25% ethanol *v/v* and 10 µs pulse duration). Research also demonstrates a significant influence of UAE on phenolic liberation from olive leaves [7].

Although the recent findings show great potential for optimal extraction of olive leaves, there is no single processing benchmark for a sustainable extraction system. This is partly because the suitability of an ideal design relies heavily on the nature/concentration of extraction solvents, solid-to-solvent ratio, extraction time/temperature, plant origin/cultivar, and physicochemical characteristics of bio-phenols, together with others.

Among the green technologies is the ohmic heating which has been viewed as a competitively ideal approach for the eco-extraction of phytonutrients from various food matrices [8,9]. Ohmic heating (Joule heating or electrical resistance heating) mainly refers to the conversion of electrical energy to heat energy in foods. This phenomenon comes about when the electric current travels through the interior of the food (that is, resistant to electric flow) which, in turn, by the effect of their electric resistance, results in the temperature increase within the product [8,10,11]. The favorable features of ohmic heating primarily include (i) exertion of internally uniform heat supply within the food, in a short period of time, and (ii) low energy/running cost [9,12]. The potential effectiveness of this method developed the idea of performing this research to examine its feasibility on olive leaf extraction. In this respect, the purpose of this study was to evaluate the effect of ohmic extraction, compared to the conventional heating and *Control* (solvent) methods, on the extraction yield, total polyphenol content, and antioxidant activity of olive mill leaves (using different concentrations of aqueous ethanol at different temperatures). Until now, no published report has examined the efficacy of the ohmic system on the extraction of olive leaves. The present research lays down the preliminary basis for expanding the knowledge of the ohmic technology to potentially enable sustainable re-utilization and valorization of olive leaf residues.

## 2. Materials and Methods

### 2.1. Plant Materials and Chemicals

Olive mill leaves of "Picual" cultivar, were kindly supplied by "Center for Advanced Studies in Energy and Environment", University of Jaén, Campus of Las Lagunillas, Jaén, Spain. The trees, within the same age range, 40–60 yr, were managed under the same agricultural condition. The leaves were delivered to the University of Minho, Portugal, manually cleaned, washed, dried, and ground. The dry leaves (with 3.6% moisture content) were stored between 0 °C to 4 °C prior to the experiments.

The following chemicals (of analytical grade) were purchased from Sigma-Aldrich (Saint Louis, MO, USA): ethanol (99.8%), sulfuric acid (≥95%), 2,2-diphenyl-1-picrylhydrazyl (DPPH), Folin–Ciocalteu, anhydrous gallic acid (≥98.0%), anhydrous sodium carbonate (≥99%), hydrochloric acid, D-(+)-Glucose (≥99.5%), potassium persulfate (≥99%), 2,2′-Azino-bis-3-ethylbenzothiazoline-6-sulfonic acid (ABTS), and (±)-6-Hydroxy-2,5,7,8-tetramethylchromane-2-carboxylic acid (Trolox).

### 2.2. Experimental Design

In this study, primarily, the efficacy of ohmic technique on the extraction of olive mill leaves was investigated. As shown in Figure 1, the leaves were initially cleaned (to eliminate stems/foreign objects), washed, and dried at 37 °C for 48 h. The dried leaves were size reduced with a grinder to pass through a 0.3 mm mesh and vacuum packed in polypropylene bags and refrigerated (0–4 °C) for a maximum of two weeks. The dry leaves, prior to the extraction study, were initially examined for proximate analysis.

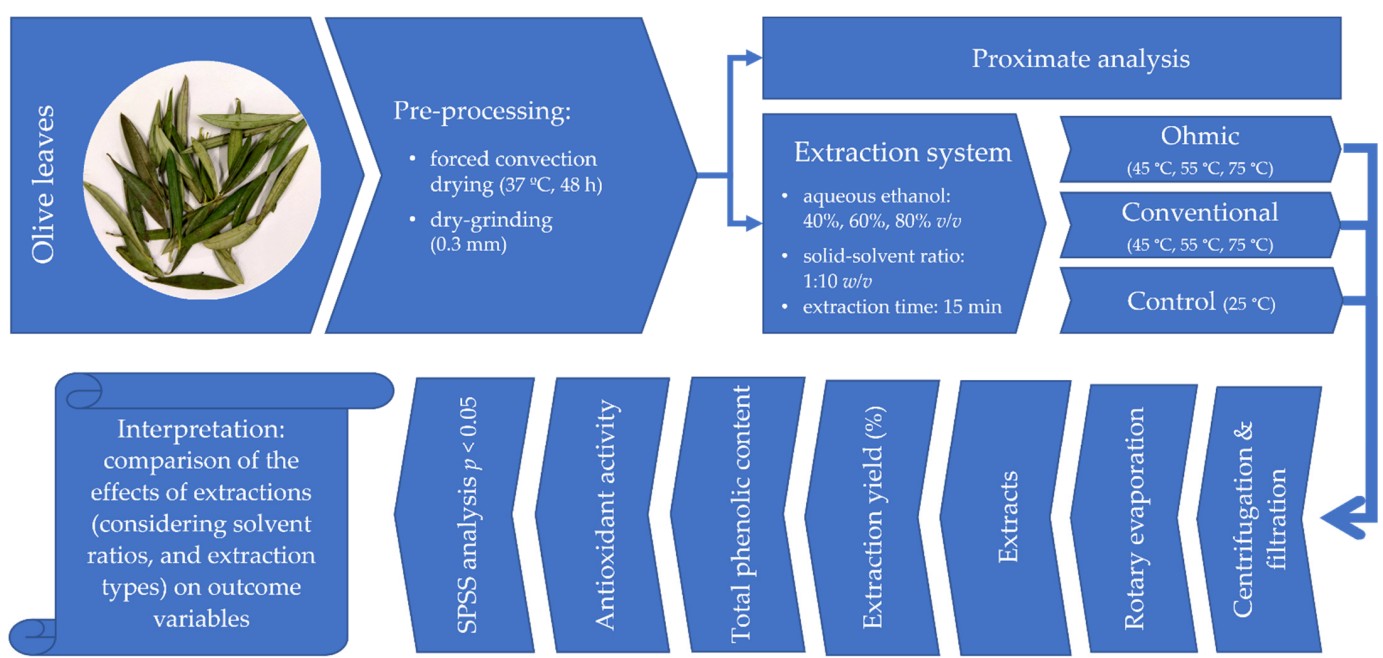

**Figure 1.** Summary of the experimental design.

*2.3. Proximate Analysis*

The dried/ground leaves were initially assessed for proximate composition as follows:

### 2.3.1. Moisture Content

Moisture content (%) of dried leaves was determined thermogravimetrically using an electronic moisture analyzer Radwag®, MAC 50/1/NH (Radom, Poland).

### 2.3.2. Total Ash

Dry ashing was carried out according to the official method of Association of Official Analytical Chemists [13]. The total ash content (%) of dry leaves was determined using the following equation:

$$\text{Ash content (\%)} = \frac{\text{Weight of Ash (g)}}{\text{Initial weight of dry sample (g)}} \times 100 \tag{1}$$

### 2.3.3. Total Fat

Crude fat was determined using a Soxtec automated extraction system (Foss Soxtec™ 8000, Hilleroed, Denmark), a Randall adaptation of Soxhlet approach [14,15], and calculated using the following equation:

$$\text{Total fat (\%)} = \frac{\text{Weight of fat in sample (g)}}{\text{Initial weight of dry sample (g)}} \times 100 \tag{2}$$

### 2.3.4. Crude Protein

Total protein was determined by means of an automated Kjeldahl analyzer (FOSS Kjeltec™ 8400, Hilleroed, Denmark). Briefly, an aliquot (1 g) of dry leaves was transferred into a Kjeldahl flask (containing two copper catalyst tablets) and heated using sulfuric acid (15 mL) to complete the digestion process. The nitrogen content (%) was automatically calculated (in accordance with the consumption of volumetric standard solution). The total protein content was calculated using the conversion factor of 6.25.

### 2.3.5. Crude Fiber

Crude fiber was determined according to the official method of AOAC [13]. Into a conical flask containing 200 mL of 0.128 M sulfuric acid, sample (2 g) was added and boiled for 30 min with periodic agitations. The solution was filtered into a discard conical flask though a muslin cloth. The filtered solid (residue) was washed into another conical flask with 200 mL base solution (0.313 M sodium hydroxide) and processed for boiling basic solution (30 min) and filtration. The filtered solid was then collected into a dry crucible, and, after drying the fiber in hot-air oven at 130 °C for 2 h, the crucible with dried fiber was weighed and incinerated in muffle furnace at 550 °C for 2 h. The weight of crucible with ash was recorded and the concentration of crude fiber (%) was calculated using the following equation:

$$\text{Crude fibre } (\%) = \frac{W_1 - W_2}{W_s} \times 100 \qquad (3)$$

where, $W_1$ = weight of crucible with dried fiber, $W_2$ = weight of crucible with ash, $W_s$ = weight of sample (g).

### 2.3.6. Total Carbohydrate

Total carbohydrate was determined using anthrone method [16]. Briefly, around 100 mg of sample was hydrolyzed in 5 mL of hydrochloric acid (2.5 N) in a boiling water bath for 3 h. After cooling at room temperature, the samples were centrifuged at 8000 rpm for 3 min. The supernatant was transferred into allocated test tubes (for sample analysis). The working standard was prepared by diluting 10 mL of stock solution (standard glucose) to 100 mL with distilled water. A series of Standards were prepared in different concentrations by pipetting 0 (blank), 0.2, 0.4, 0.6, 0.8, and 1 mL into designated test tubes and made the final volume of 1 mL with distilled water. Into each test tube (samples and standards), 4 mL anthrone reagent (containing 100 mL ice-cold concentrated sulfuric acid and 200 mg anthrone) was added and boiled for 8 min. The absorbance reading was measured at 630 nm. The carbohydrate concentration was calculated using the standard calibration curve (plotting the concentration of glucose concentration versus absorbance).

### 2.4. Extraction System

The extraction of olive leaves was assessed through: (i) ohmic heating (*OH*), (ii) conventional heating (*Conven*), and (iii) *Control* (solvent) methods. *Instrumental setup*—the main components of a bench-scale ohmic heater used in this study [Figure 2] were: (i) a Pyrex glass reactor chamber (10 cm height, 9 cm i.d, with 100 mL capacity) equipped with two titanium electrodes (with 5 cm distance in between), and a K-type thermocouple to be positioned in the center, (ii) a function generator (Protek 2MHz Sweep Function Generator, Long Branch, NJ, USA) enabling waveform adjustment, (iii) an amplifier from which the electrodes received signals, (iv) a hand-held oscilloscope (Industrial ScopeMeter® 125/S 40 MHz, Fluke, Everett, WA, USA), and (v) a data acquisition system (LabVIEW 7 Express system software, Champaign, IL, USA) to monitor the temperatures of the food received from the thermocouple sensor.

*Ohmic Extraction*—five grams of sample (dried/ground leaves) mixed with 50 mL of the selected concentration of aqueous ethanol (40%, 60%, and 80% EtOH, *v/v*) and transferred into the reactor chamber and processed for the extraction, using the selected extraction temperature (45 °C, 55 °C, and 75 °C) for 15 min. The homogeneity of sample solution and uniformity of heat transfer during the extraction process were achieved by means of a magnetic stirrer positioned in the reactor chamber stirred at 150 rpm. The temperature variations in the sample were measured with a thermocouple, equipped with a data logger operating through a software computer system for data acquisition. The hydroethanolic extract was centrifuged at 5000 rpm for 15 min. The supernatant was filtered through a Whatman filter paper. The filtrate was then subjected to rotary evaporation at 40 °C for 1 h. The extract was nitrogen flushed, transferred to an amber glass bottle and stored at −20 °C until analytical experiments.

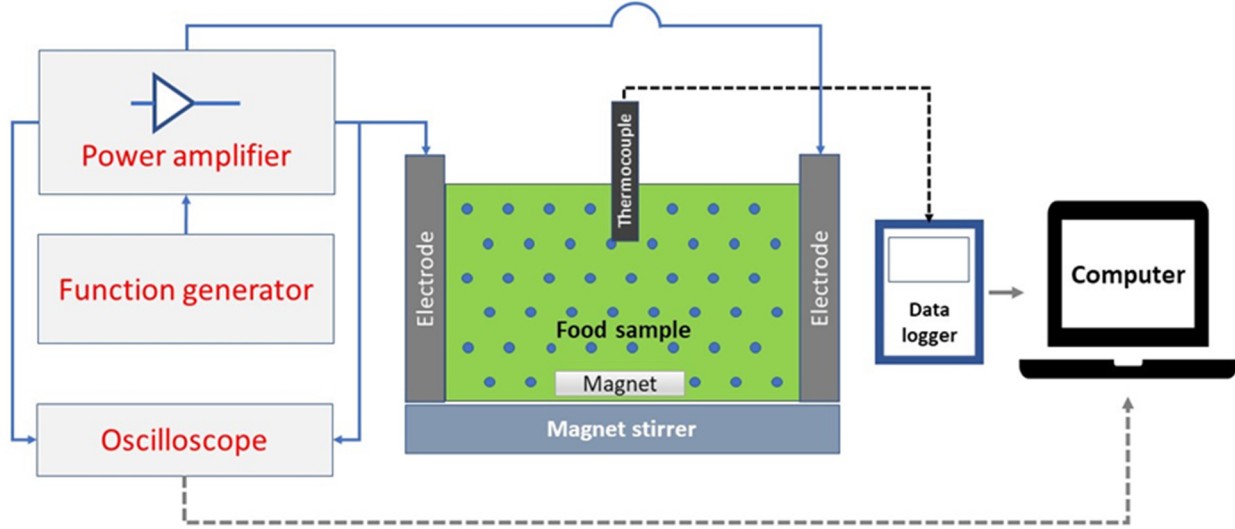

**Figure 2.** Layout of ohmic heating system used for olive leaf extraction.

*Conventional Heating Extraction*—to effectively assess the influence of the ohmic method, a conventional thermal extraction was performed, under the same conditions/system components applied for the ohmic with the exception that in the glass reactor (containing the sample) only thermocouple was used without using electrodes. The system was equipped with a circulating/thermostat water bath enabling thermal/temperature control.

*Control (solvent) extraction*—in addition to the abovementioned methods, a solvent extraction was carried out as *Control* through the same solvent conditions with the exception that no heat treatment was applied (using only agitation at room temperature, 25 °C).

### 2.5. Extraction Yeild

The extraction yield (%), based on thermal gravimetric analysis, was evaluated through drying of 1 mL aliquot of extract at 105 °C and the weight measurement was monitored in two-hour intervals until it reached equilibrium point/constant weight. The results were calculated as follows and expressed as gram extract per 100 g of dry leaves:

$$\text{Yield of extracted leaves (\%)} = \frac{W_e}{W_i} \times 100 \qquad (4)$$

where, $W_i$ = Initial weight of dry sample (g), and $W_e$ = Weight of dry extract (g).

### 2.6. Total Phenolic Content (TPC)

The gross quantification of total phenolics was determined using Folin–Ciocalteau assay, described by Singleton et al. [17] with some modifications. Into a 10 mL volumetric flask, 100 μL of extract, 6 mL distilled water and 500 μL of Folin–Ciocalteau reagent were added. The mixture was vortex mixed and allowed to stand at 25 °C for 8 min before adding 1.5 mL of 20% sodium carbonate. The mixture was made up to 10 mL final volume with distilled water and incubated in dark/cool place for 60 min. The absorbance was measured at 760 nm using UV-vis spectrophotometer. Total phenolic content was calculated against gallic acid standard curve and reported as mg of gallic acid equivalents/g extract dry weight (mg GAE/g extract d.w.).

### 2.7. Antioxidant Activity

2.7.1. ABTS Radical Scavenging Activity

The ability of bioactive compounds in extracts to inhibit ABTS radical activities was assessed using the method described by Re et al. [18] with slight modifications. The radical cation (ABTS$^{\bullet+}$) was liberated through the reaction of ABTS stock solution (7 mM) with

potassium persulfate (2.45 mM). After incubating in dark for 16 h, the $ABTS^{\bullet+}$ solution was diluted with ethanol to an absorbance of 0.70 at 734 nm. An aliquot of sample (properly diluted) was then added to the solution (1:10 $v/v$), mixed, and the absorbance was measured at 734 nm. The absorbance readings were calculated against Trolox calibration curve and expressed as mM Trolox equivalents/g extract (mM TE/g extract d.w.).

### 2.7.2. DPPH Radical Scavenging Activity

The DPPH radical scavenging assay was carried out following the method of Brand-Williams et al. [19] with slight modifications. An aliquot of 3.9 mL DPPH working solution (36 µg/mL ethanol) and 100 µL of extract (properly diluted) were added into a test tube and made up to volume (10 mL) with ethanol and vortex mixed for 10 s. The mixture was incubated in the dark/room temperature for 60 min. The absorbance readings (517 nm) were calculated against Trolox standard curve and the results were expressed as mM TE/g extract d.w. Additionally, the following equation was used to determine the percent inhibition of radical activity:

$$\% \text{ Inhibition of DPPH radical} = \frac{A_0 - A_1}{A_0} \times 100 \tag{5}$$

where, $A_0$ = Absorbance of control (DPPH solution without extract), and $A_1$ = Absorbance of sample (DPPH solution mixed with extract).

### 2.8. Statistical Analysis

Each experiment was carried out in triplicate and the results were recorded as the mean values $\pm$ SD. The significant differences ($p < 0.05$) between the mean values ($\pm$SD) of all determinations were statistically assessed via Analysis of Variance (ANOVA) using SPSS software, version 27.0. A difference was considered statistically significant when $p < 0.05$. The quality parameters (response variables) of the extracts were individually analyzed using factorial ANOVA (two-way ANOVA) to assess the interactive effects of two independent variables (solvent ratio and extraction method) and their joint effects on the mean values of dependent variables individually. The assumption of homogeneity of equal variance was assessed through the Levene's test (homogeneity of variance assumption was satisfied/not violated when $p$-value was greater than 0.05).

### 3. Results and Discussion

The selection of the extraction system plays a significant role in the overall extraction yield and quantity/functionality of recovered bioactive compounds. Food and nutraceutical producers require a viable/affordable extraction design to sustainably reuse/valorize the byproduct streams, including olive leaves. A range of methodologies are being researched, encompassing both innovative and conventional methods. While being highly effective, various extraction methods represent variabilities in phenolic quantities/bioactivities; reflecting their variations in diffusivity, solubility, polarity, and heat sensitivity, along with others. In this regard, the extent of phenolic recovery/bioactivity rests highly on the selected extraction design/parameters that may favorably/unfavorably affect the food microstructure and the solute mass transfer.

In the present study, the investigation into the efficacy of ohmic application, using varied solvent concentrations and extraction temperatures, was carried out for olive leave extraction, with special attention to (i) extraction yield, (ii) total phenolic content, and (iii) antioxidant activity of extracts. It is noteworthy that ethanol, as a preferred extraction solvent, is considered a green/biodegradable solvent with minimum toxicity (typically produced in the course of fermentation of sugars from plants/algae) and its ability to favorably extract phenolics from olive leaves has been substantiated in recent studies.

### 3.1. Proximate Analysis

The selected macromolecules present in dry/ground leaves were determined to evaluate their main elemental characteristics. The proximate composition of olive leaves (g/100 g d.w.) was examined in terms of moisture, ash, total fat, crude fiber, crude protein, and carbohydrates (Table 1).

**Table 1.** Proximate composition (mean ± SD) of dried/ground olive leaves.

| Component g/100 g Dry Leaves | Mean ± SD |
|---|---|
| Moisture | 3.57 ± 0.18 |
| Total ash | 10.82 ± 0.8 |
| Crude fat | 4.13 ± 0.02 |
| Crude protein | 8.02 ± 0.13 |
| Crude fiber | 35.41 ± 0.35 |
| Carbohydrate | 37.65 ± 1.30 |

The dry ash, that is the inorganic residue remaining after combustion, represents the mineral content in the food. It is routinely performed as part of the characterization of chemical/nutritional attributes of the food of interest. The ash content measured in this experiment (10.82%) is relatively close to that reported by Contreras et al. [20], where the olive mill leaves (from the same cultivar/growing region used in this study) exhibited around 10.04%. Doménech et al. [21] also experimented on olive leaves from the same growing region and found 9.1% ash in the raw biomass. The concentration of ash is partly determined by (i) pre-harvest/agricultural conditions, (ii) sample origins/collection methods, and (iii) pre-processing approaches, such as blanching and size reduction, together with others. Caballero et al. [22] reported 8.22% ash in Spanish olive leaves, obtained from the olive mill (pneumatically separated from olive drupes, washed, dried, and ground). Zeitoun et al. [23] explored 5.80%, 5.77%, and 4.58% ash content in blanched/dried, solar-dried, and oven-dried leaves, respectively. Further, Cavalheiro et al. [24], through their experiment, found 4.65%, 6.00%, 4.37%, 4.85%, and 5.36% ash in tree-harvested leaves from Arbosana, Ascolano, Grappolo, Koroneiki, and Negrinha do Freixó, respectively.

As shown in Table 1, the dry leaves in this study contained 8.02 ± 0.13 g/100 g crude protein, which is as close as that reported by Contreras et al. [20], where 8.10% protein was found in olive mill leaves from the same cultivar/region. The protein content in olive leaves varies among different studies; partly due to the variations in growing regions and soil fertility, among others. Examples are 4.95%, [23], 5.45% [25], 6.9%–8.1% [26], 10.6% [27], and 7.8% [21].

The fat content determined in this study was 4.13 ± 0.02 g/100 dry leaves. There is also a broad spectrum of the reported total lipids in olive leaves. Examples are 2.29% [28], 6.54% [25], 9.9% [26], and 9.13%–9.80% [24]. One of the main deciding factors affecting the fat content, beyond the cultivar variations and growing regions, is the particle size of samples, which is dependent on the selected milling method. The dry leaves, ground to pass through a 60-mesh screen, reportedly contained 7.9% total lipids [27]. In another study, the pulverized leaves with 1 mm particle size contained 3.21% total fat [29].

Total carbohydrates constituted around 37.65% in this study. Among the published reports include 36.75% and 45.96% (in oven-dried and solar-dried leaves, respectively) [23]; 29.20% and 25.4% (in non-blanched and blanched leaves, respectively) [30]; and 27.58% [25]. Moreover, focusing on different varieties, Cavalheiro et al. [24] obtained 8.74%, 11.60%, 12.75%, 16.70%, and 32.63% in olive leaves from Negrinha do Freixó, Koroneiki, Ascolano, Arbosana, Grappolo cultivars, respectively.

The crude fiber accounted for 35.41% in the leaves of this study. Examples in previous studies are: 14.5% in oven-dried/60-mesh powdered leaves [27]; 32.83% in oven-dried leaves freshly harvested from trees [28]; 7.0% in heat-pump-dried leaves [25]; and 4.71% in oven-dried leaves [23].

### 3.2. Extraction Yield

The yield percentage of the residual dry matter (the extractable matter) was determined though convection drying. The extraction ability of ohmic heating was examined compared to *Conven* and *Control*. The mean values of the extraction yield (%) obtained from different extraction methods are shown in Table 2.

**Table 2.** Mean values (±SD) of yield and functional properties of extracted olive mill leaves – extraction yield (g/100g dry leaves), total phenolic content (mg GAE/g extract), TEAC assays (ABTS and DPPH radical scavenging activity mM TE/g extract), and % inhibition of DPPH and ABTS radicals.

| Extraction System | | Extraction Yield (g/100 g Dry Leaves) | TPC (mg GAE/g Extract) | Antioxidant Activity | | | |
|---|---|---|---|---|---|---|---|
| Method | % EtOH (v/v) | | | ABTS$^{\bullet+}$ Inhibition (mM TE/g Extract) | ABTS$^{\bullet+}$ Inhibition (%) | DPPH$^{\bullet}$ Inhibition (mM TE/g Extract) | DPPH$^{\bullet}$ Inhibition (%) |
| *OH* 45 °C | 40 | 22.02 ± 0.15 | 33.73 ± 0.21 | 0.44 ± 0.01 | 67.06 ± 1.03 | 0.93 ± 0.07 | 85.23 ± 0.10 |
| *OH* 45 °C | 60 | 28.30 ± 0.12 | 38.37 ± 0.32 | 0.45 ± 0.02 | 68.93 ± 0.18 | 0.96 ± 0.01 | 86.67 ± 0.19 |
| *OH* 45 °C | 80 | 30.80 ± 0.11 | 30.45 ± 0.39 | 0.49 ± 0.10 | 69.60 ± 0.41 | 1.08 ± 0.05 | 87.70 ± 0.25 |
| *OH* 55 °C | 40 | 23.21 ± 0.15 | 34.36 ± 0.36 | 0.44 ± 0.10 | 70.45 ± 0.47 | 0.93 ± 0.10 | 89.94 ± 0.13 |
| *OH* 55 °C | 60 | 30.20 ± 0.14 | 42.53 ± 0.31 | 0.49 ± 0.11 | 74.72 ± 0.45 | 1.01 ± 0.06 | 90.77 ± 0.52 |
| *OH* 55 °C | 80 | 31.10 ± 0.17 | 31.63 ± 0.43 | 0.55 ± 0.02 | 77.56 ± 0.36 | 1.15 ± 0.01 | 92.55 ± 0.12 |
| *OH* 75 °C | 40 | 27.53 ± 0.13 | 34.06 ± 0.23 | 0.48 ± 0.01 | 73.95 ± 0.39 | 1.04 ± 0.50 | 90.85 ± 0.44 |
| *OH* 75 °C | 60 | 28.50 ± 0.12 | 41.13 ± 0.40 | 0.54 ± 0.05 | 76.79 ± 0.17 | 1.11 ± 0.30 | 91.56 ± 0.56 |
| *OH* 75 °C | 80 | 34.53 ± 0.41 | 30.23 ± 0.35 | 0.62 ± 0.15 | 78.72 ± 0.48 | 1.21 ± 0.04 | 92.80 ± 0.57 |
| *Conven* 45 °C | 40 | 19.41 ± 0.54 | 23.92 ± 0.16 | 0.44 ± 0.02 | 67.08 ± 0.08 | 0.96 ± 0.04 | 86.54 ± 0.07 |
| *Conven* 45 °C | 60 | 21.39 ± 0.55 | 28.44 ± 0.31 | 0.47 ± 0.03 | 68.60 ± 0.35 | 1.05 ± 0.01 | 88.33 ± 0.13 |
| *Conven* 45 °C | 80 | 21.18 ± 0.27 | 26.75 ± 0.32 | 0.48 ± 0.02 | 70.10 ± 0.94 | 0.98 ± 0.03 | 87.21 ± 0.14 |
| *Conven* 55 °C | 40 | 19.63 ± 0.16 | 23.48 ± 0.29 | 0.45 ± 0.17 | 73.99 ± 0.22 | 1.03 ± 0.03 | 89.40 ± 0.31 |
| *Conven* 55 °C | 60 | 21.40 ± 0.12 | 32.86 ± 0.41 | 0.49 ± 0.05 | 75.78 ± 0.39 | 1.09 ± 0.06 | 90.31 ± 0.13 |
| *Conven* 55 °C | 80 | 22.52 ± 0.54 | 24.67 ± 0.28 | 0.48 ± 0.03 | 74.79 ± 0.20 | 1.00 ± 0.11 | 89.10 ± 0.09 |
| *Conven* 75 °C | 40 | 19.34 ± 0.49 | 24.75 ± 0.18 | 0.47 ± 0.18 | 74.34 ± 0.27 | 0.98 ± 0.01 | 88.91 ± 0.08 |
| *Conven* 75 °C | 60 | 22.40 ± 0.13 | 31.56 ± 0.20 | 0.52 ± 0.10 | 76.07 ± 0.10 | 1.10 ± 0.03 | 91.39 ± 0.19 |
| *Conven* 75 °C | 80 | 22.20 ± 0.42 | 28.94 ± 0.30 | 0.49 ± 0.09 | 75.66 ± 0.30 | 1.04 ± 0.01 | 90.40 ± 0.20 |
| *Control* 25 °C | 40 | 19.47 ± 0.41 | 19.75 ± 0.28 | 0.27 ± 0.18 | 59.07 ± 1.12 | 0.65 ± 0.07 | 75.49 ± 0.19 |
| *Control* 25 °C | 60 | 20.29 ± 0.43 | 25.26 ± 0.23 | 0.32 ± 0.01 | 65.74 ± 0.32 | 0.69 ± 0.12 | 76.54 ± 0.11 |
| *Control* 25 °C | 80 | 19.90 ± 0.20 | 23.25 ± 0.19 | 0.30 ± 0.01 | 63.88 ± 0.10 | 0.73 ± 0.01 | 78.06 ± 0.14 |

The *OH* represents ohmic extraction using different concentrations of aqueous ethanol (40%, 60%, and 80% EtOH *v/v*) at different temperatures (45 °C, 55 °C, and 75 °C) for 15 min. The *Conven* represents groups of samples with conventional heating extraction using the same solvent ratios/temperatures as those in *OH* system. The *Control* represents extraction at room temperature using the same abovementioned solvent concentrations. TEAC, Trolox equivalent antioxidant capacity; TPC, total phenolic content (mg gallic acid equivalents/g extract d.w.).

The results from the post-hoc test (comparing different mean values of the extraction yield between different groups of extraction methods) showed that the extent of difference was significantly great between the *OH* groups (particularly 75 °C) and other groups of extraction methods. The difference was more pronounced between *Control* and *OH* groups ($p < 0.001$). As illustrated in Figure 3, the extraction solvent with 80% ethanol showed the highest values of yield across all groups of extractions (except for *Control* and *Conven*, 75 °C and 45 °C), particularly in *OH* groups exhibiting 34.53%, 31.10%, and 30.80% with 75 °C, 55 °C, and 45 °C, respectively.

The statistical significance of interactions between the two main factors (where the extraction methods were individually compared vs. solvent ratios) was studied through a pairwise comparison, which confirmed whether/not the extraction methods are significantly dependent on the solvent concentrations ($p < 0.05$). It was evident that the mean differences between the solvent ratios in *OH* groups were highly significant ($p < 0.001$), compared to *Conven* and *Control*. In other words, the reliance of ethanol concentration was greatest in *OH* groups that showed the highest effects on the extraction yield ($p < 0.001$). It was also found that in *Control* (between 40% and 80%), *Conven* 45 °C (60% and 80%), and *Conven* 75 °C (60% and 80%), the mean differences were not statistically significant ($p > 0.05$).

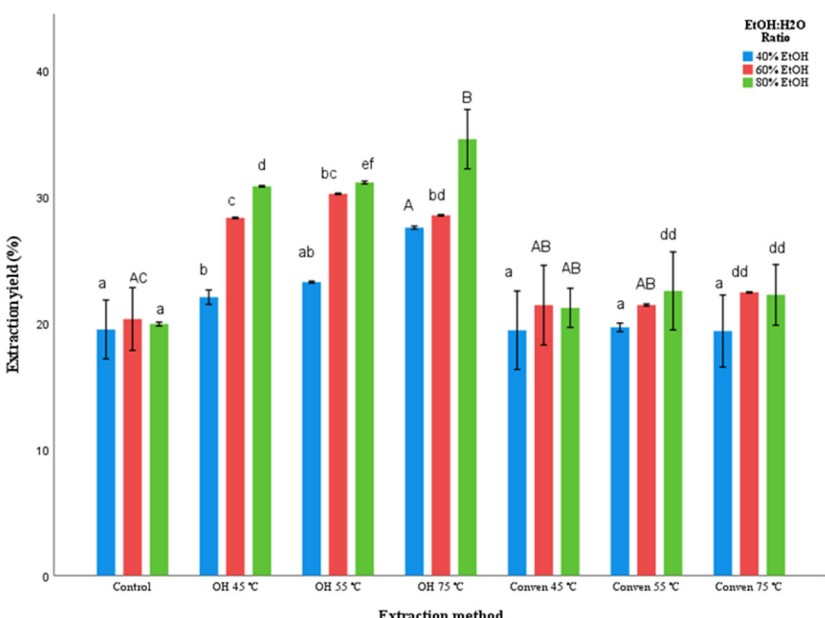

**Figure 3.** Extraction yield (g extract/100 g dry olive leaves) obtained through various extraction methods. The *Control* represents the extraction with no heat treatment (25 °C). The *OH* represents ohmic extraction using different concentrations of aqueous ethanol (40%, 60%, and 80% EtOH *v/v*) and different temperatures (45 °C, 55 °C, and 75 °C), The *Conven* represents groups of samples with conventional heating extraction using the same solvent ratios/temperatures as those employed for *OH* samples. Mean values of % yield are shown with standard deviation error bars for each category of extraction method. Different letters (a–f, A–C) above the bars indicate statistically significant differences between means ($p < 0.05$).

The extraction yield of olive leaves has been researched in numerous studies. Şahin et al. [31] examined different solvent proportions within a selected temperature/time range. Among the main findings, the extraction yields (50 °C for 60 min) accounted for 322.33 mg/g and 328.82 mg/g using 50% ethanol and 100% methanol, respectively. In the study of Lama-Muñoz et al. [32], using 80% ethanol with 1:6 solid/solvent ratio, the highest extraction yield (dry residue of leave biomass) was around 27.55%. Further, among the key results reported by Doménech et al. [21], total extraction yield (%) of olive leaves, using water, was found to be around 35.0%. The rate of extraction yield is greatly reliant on the extraction methods/condition. However other contributing factors may decidedly affect the increase/decrease in extraction yield, which include variations in sample collections among different agro-industrial practices, cultivars, geographical origin, tree life time, storage conditions, etc.

### 3.3. Total Phenolic Content (TPC)

The concentration of polyphenols is among the key factors responsible for antioxidant activity/capacity in food. In this study, TPC was measured through the reaction of Folin–Ciocalteu reagent with total phenols available in the extracts. Figure 4, illustrates the mean values of the TPC measured for all extraction groups (mg GAE/g extract d.w.). In respect of extraction methods, ohmic heating represented the highest values, notably at 55 °C, subsequently with 75 °C and 45 °C. Samples extracted with *OH* 55 °C yielded 42.53, 34.36, and 31.63 mg GAE/g extract using 60%, 40%, and 80% ethanol, respectively.

Overall, 60% ethanol exerted the greatest effect on increased TPC in all extraction groups. The magnitude of difference between 60% and 80% (in *OH* groups), and 60% and 40% (in *Conven* and *Control* groups) was significantly large ($p < 0.001$).

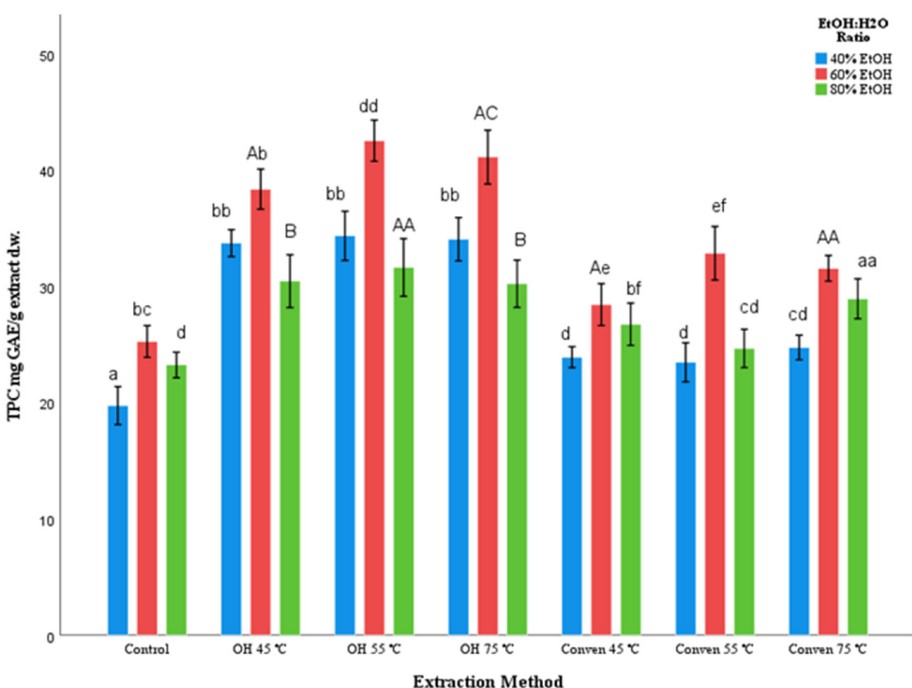

**Figure 4.** Total phenolic content (TPC) mg GAE/g extract d.w. through various extraction methods are presented with standard deviation bars. The *Control* represents the extraction with no heat treatment (25 °C). The *OH* represents ohmic extraction using different concentrations of aqueous ethanol (40%, 60%, and 80% EtOH *v/v*) and different temperatures (45 °C, 55 °C, and 75 °C). The *Conven* represents groups of samples with conventional heating using the same solvent ratios/temperatures as those applied for ohmic system. Different letters (a–f, A–C) above the bars indicate statistically significant differences between means ($p < 0.05$).

In previous research experiments, using different extraction techniques/designs, various concentrations of TPC have been reported. Goldsmith et al. [33] obtained 32.4 mg GAE/g using aqueous extraction at 90 °C for 70 min, and 1:60 g/mL solid–solvent ratio. In their study, Lee et al. [34] detected that olive leaves extracted with 80% ethanol contained phenolics around 148 mg/g tannic acid equivalents. Research unveiled the potential of the optimized UAE (43.61% ethanol for 59.99 min at 34.18 °C) for increased phenolic recovery up to 43.825 mg GAE/g dry leaves [35]. Sánchez-Gutiérrez et al. [36] found higher total phenolics (76.1 mg GAE/g d.w.) in Soxhlet-extracted leaves compared to those extracted by MAE with 80% ethanol for 10 min (54.0 mg GAE/g d.w.). On the other hand, Da Rosa [3] described that the optimized MAE (using water at 86 °C for 3 min) is more effective, compared to the maceration, in the increase in phenolic liberation (by 82%). Another study demonstrated that olive leaves (dried at 60 °C for 120 min), through supercritical extraction, contained 36.1 mg GAE/g dry leaf [37].

The performance of extraction solvents in diffusivity may vary due to the phenolic complexity (in respect of solubility and polarity), and this constitutes a challenge to select the appropriate solvent(s) to ensure maximum extraction of bio-phenols while maintaining their bio-functionalities. It may not be ideally practical to use 100% of a single solvent and the use of excessive polar or non-polar solvents may work poorly on the release of phenols. Further, the choice of solvents comes with a challenge of choosing green/non-toxic ones, such as water and ethanol. There is much research investigating the effects of solvent nature/ratios on phenolic recovery from olive leaves. Şahin et al. [31] conducted research on various solvents/solvent concentrations, together with considering extraction temperature/time. Among the main findings, the samples with 100% methanolic extracts (50 °C for 60 min) yielded around 328.82 mg/g extract that was greater than those obtained by 100% ethanol (176.42 mg/g extract) under the same time/temperature conditions. However,

the proportional use of ethanol showed improved extraction ability, as using 50% ethanol yielded 322.33 mg/g extract under the same temperature/time conditions. In another study, the extraction of olive leaves (from Koroneiki cultivar) with 50% ethanol, using maceration and microwave-assisted extractions, yielded phenolics of around 69.027 mg TAE/g and 88.298 mg TAE/g d.w., respectively [38].

Other crucial factors affecting polyphenol content include pre-processing approaches, such as blanching, drying, and grinding methods. Ahmad-Qasem et al. [39] observed that hot-air drying at higher temperature for shorter drying time (120 °C for 12 min) enabled increased phenolic content (59 mg GAE/g d.w.), compared to that obtained through lower temperatures for a longer time (45 mg GAE/g d.w., using 70 °C for 50 min). Research also demonstrated that drying olive leaves (Chemlali cultivar) with an infrared drying method at 70 °C and 40 °C exhibited around 5.14 and 2.13 g caffeic acid/100 g d.w., respectively [40].

### 3.4. Antioxidant Activity

The endogenous polyphenols in olive leaves potentially exert antioxidant activity via deactivating/stabilizing free radicals. The following experiments enabled identification of the protective ability of the extracts against oxidative damage of free radicals.

### 3.4.1. Trolox Equivalent Antioxidant Capacity (TEAC)

The free radical scavenging activity of extracts was determined using two different *in vitro* antioxidant assays (DPPH and ABTS). The method relies on the electron-transfer mechanism, that is, the potential of antioxidants to inhibit radical activities via transferring electrons.

As shown in Figure 5, the *OH* groups showed relatively similar pattern in both DPPH and ABTS assays (exhibiting the highest values with 80%, and the lowest values with 40% ethanol). The different patterns between them were as follows: (i) *Control* groups –in DPPH, the 80% exhibited the highest value (0.73 mM TE/g), while, in ABTS, the 60% showed the highest value (0.32 mM TE/g), (ii) *Conven* 45 °C—where 60% represented the highest in DPPH (1.05 mM TE/g), and the 80% showed the highest in ABTS (0.48 mM TE/g), and (iii) *Conven* 55 °C—antioxidant capacity in descending order: 60%, 40%, and 80% in DPPH results, and 60%, 80%, and 40% in ABTS results.

The mean values of antiradical activity detected by DPPH assay were evidently greater than the corresponding values identified by ABTS method. Results confirmed, from both assays, that ohmic heating represented the highest antioxidant potency compared to *Conven* and *Control*, particularly in *OH* 75 °C, ranging from 1.21 to 1.04 mM TE/g extract (with DPPH), and 0.62 to 0.48 mM TE/g extract d.w. (with ABTS).

In previous studies, the TEAC values differed largely among various processing methods/conditions, cultivar/growing regions, antioxidant assays, and sample collections (tree-picked, pruning biomass, or olive mill leaves).

Herrero et al. [41], through their experiment on the extraction of olive leaves (residues of olive oil industry, Spain), employed pressurized liquid extraction and noted that using water (200 °C) and ethanol (150 °C) exhibited 2.661 mM TE/g and 0.677 mM TE/g, respectively. In their research, Lins et al. [42] observed around 0.215 and 0.148 g TE/g extract d.w. in DPPH and ABTS, respectively. Abaza et al. [43], focusing on the effect of solvent natures on the extraction of olive leaves from Chétoui cultivar, found ABTS radical inhibiting values in a range of 629.87–1064.25 μmol TE/g d.w., with 80% methanol giving the greatest antiradical potency. Goldsmith et al. [33], through optimization of aqueous extraction, detected around 85.26 mg TE/g using the DPPH method. Nicolì et al. [44] used 60% ethanol for the extraction of fifteen varieties of Italian olive leaves. Among their findings, the DPPH radical inhibition (8.67–29.89 μmol TE/mg extract, d.w.) in leaves from Minerva and Itrana cultivars represented the lowest and the highest ranking, respectively. Also, the ABTS radical scavenging activities have been reported in (i) olive mill leaves, in a range of 18,234–25,459 μmol TE/100 g, and (ii) tree-picked olive leaves, of around 59,651 μmol TE/100 g [20]. The research of Orak et al. [45] discovered a range of TEAC

values in the selected olive leaves with different genotypes cultivated in Turkey, which ranged from 0.7 to 1.01 mM TE/g extract, with Esek Zeytini and Uslu genotypes giving the highest and the lowest ABTS scavenging effects, respectively. Moreover, in the research of Hayes et al. [46], the commercial olive leaf extract exhibited around 37.93 g TE/100 g extract d.w.

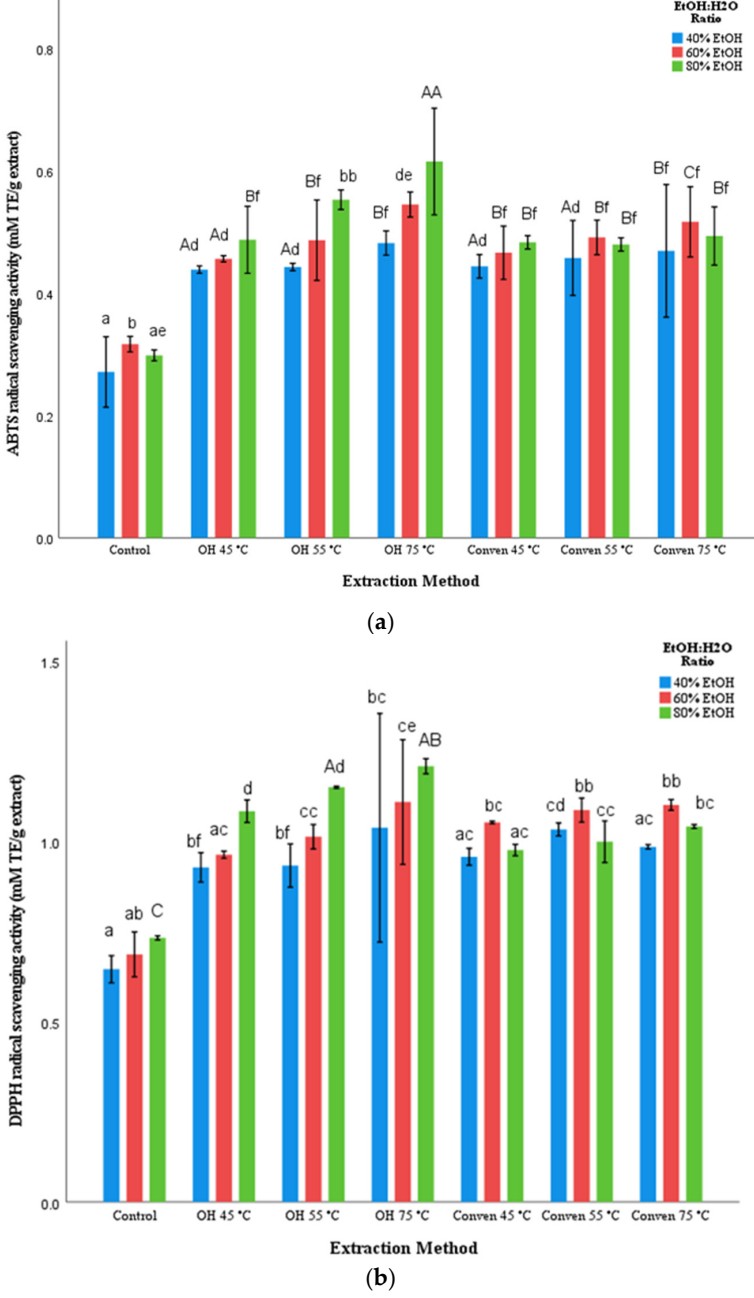

**Figure 5.** Mean values of Trolox equivalent antioxidant capacity (TEAC) in olive leaf extracts. (**a**) ABTS radical scavenging activity (mM TE/g extract d.w.). (**b**) DPPH radical scavenging activity (mM TE/g extract d.w.). Different letters (a–f, A–C) above the bars indicate statistically significant differences between means ($p < 0.05$). The Control represents the extraction with no heat treatment (25 °C). The OH represents ohmic extraction using different ratios of aqueous ethanol (40%, 60%, and 80% EtOH *v/v*) and different temperatures (45 °C, 55 °C, and 75 °C), The Conven represents groups of samples with conventional heating using the same solvent ratios/temperatures as those in OH method.

### 3.4.2. Relationship between Antioxidant Activity (%) and TPC

To present more informative data, the results were further evaluated to compare the % inhibition of free radicals with the mean values of TPC for each extraction group (Figure 6). Regardless of solvent ratios, comparatively, similar trends were seen in the results of TPC and antioxidant activity (both methods). However, the highest mean values in TPC were observed in *OH* 55 °C, while the highest values of antiradical activity in both assays belonged to *OH* 75 °C. In addition, the magnitude of mean differences of TPC between *OH* groups and *Conven* groups was much greater compared to the corresponding groups in ABTS and DPPH.

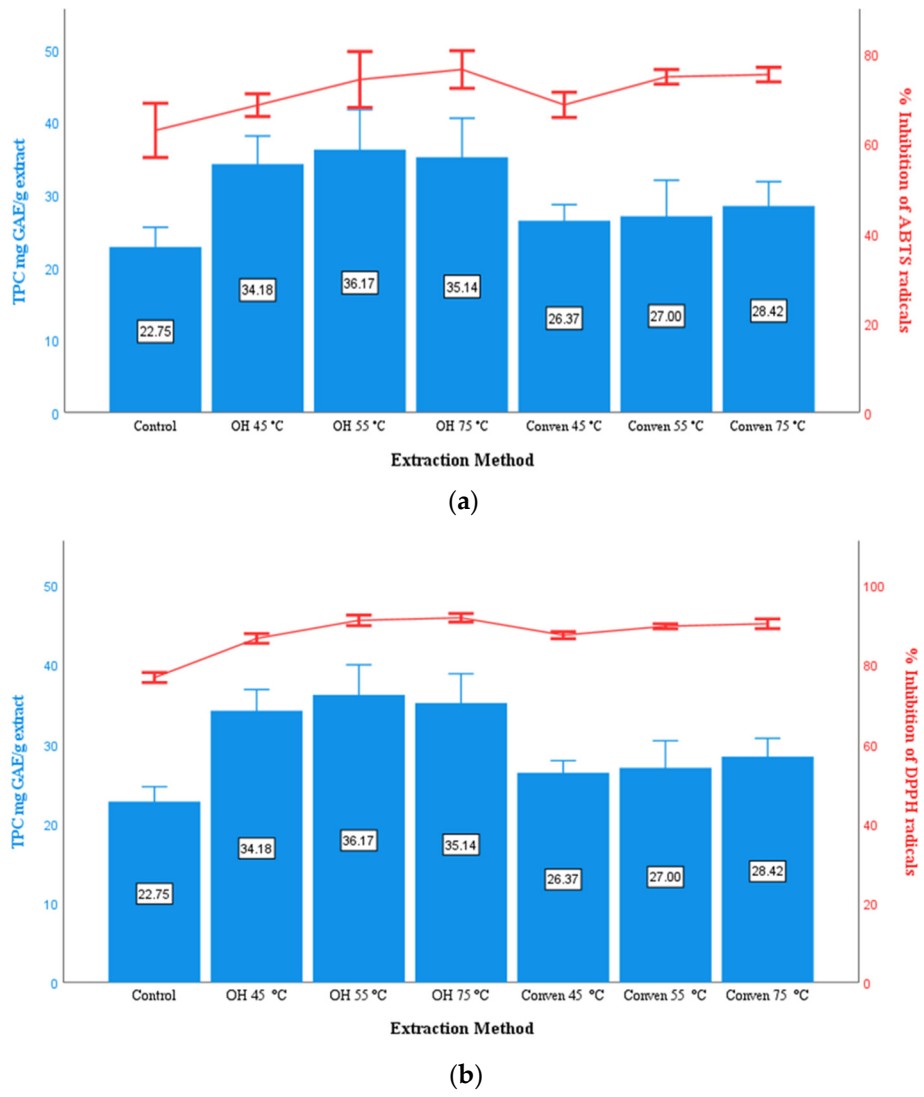

**Figure 6.** (**a**) Relationship between % ABTS radical scavenging activity and total phenolic content. (**b**) Relationship between % DPPH radical scavenging activity and total phenolic content. The values were obtained from various extraction methods (OH, Conven, and Control). Error bars represent the standard deviation (*p* < 0.05).

It was determined, overall, that the relation between total polyphenol content and antiracial activity (DPPH and ABTS) was statistically significant (*p* < 0.001). By reason of their biological activity, the greater proportion of polyphenols in plant tissues is expected to be positively correlated with higher antioxidant activity. The association between these two outcome variables in olive leaf extracts have been discussed in several studies. It is noteworthy that, due to the variability and complexity of phenolic compounds in different

olive leaves (in terms of molecular structure, polarity, solubility, and concentration), the findings may not present a unique pattern for all types of olive leaves (even from the same cultivar/growing region). In the research of Papoti et al. [47], olive leaf extracts with higher total phenolic content exhibited considerable DPPH radical scavenging capacity (%), and it was highlighted that some of the examined cultivars, such as Atsilochou, Asprolia, Chrysophilli, and Pikrolia, represented particularly significant antiradical potential, ranging from 89 to 92% and 91 to 94%, in ethanolic and methanolic extracts, respectively. Monteleone et al. [48] examined Biancolilla leaves, collected from tree pruning, and discovered that hydroalcoholic extracts yielded higher TPC, while the DPPH radical inhibiting values (%) were relatively close in all types of both aqueous and ethanolic/methanolic extracts (range: 88.90%, 90.85%, and 91.20%, using water (90 °C), 70% ethanol, and 70% methanol, respectively). Kiritsakis et al. [49] described that the use of methanol (60%) for successive extraction of different cultivars (Koroneiki, Kalamon, and Megaritiki) enabled total phenolics around 6196, 5579, and 6094 mg GAE/kg dried leaves, respectively. These authors confirmed the high association between polyphenols and DPPH radical inhibiting potency. Irakli et al. [50] explored positive correlation, particularly in the leaves extracted by the ultrasound-assisted method, using 50% acetone, 50% methanol, and 50% ethanol. Sánchez-Gutiérrez et al. [36] found positive linearity in Soxhlet-extracted leaves using 50% ethanol (around 76.1 mg GAE/g with 78.01 mg TE/g), 75% ethanol (around 71.9 mg GAE/g with 72.043 mg TE/g), and water (around 67.6 mg GAE/g with 65.765 mg TE/g d.w.).

In this study, ohmic heating proved significantly useful for greater recovery of total polyphenols and, correspondingly, higher antioxidant activity in olive leaves. This information would further benefit from extensive research assessments to compare the ohmic with a range of green/competing extraction methods (focusing on various operating/processing parameters). For example, it is of value to compare ohmic method with MAE (e.g., in terms of frequency and energy efficiency) which has been demonstrated to be competitively effective in phenolic extraction and, using less energy/extraction time, it enables uniform heat transfer from interior of the food and exerts effects on rupturing the cell walls which assists in the release/extraction of desired bio-compounds [51]. Additionally, the extraction potency of ohmic can ideally be compared with non-thermal emerging techniques, such as PEF and HVED, e.g., in respect of electric strength, distance of electrodes, and frequency. Research, investigating the effects of green technologies (HVED, PEF, and UAE) on polyphenolic extraction from blueberry pomace, highlighted the weighty influence of electric strength and energy input on the rate of permeabilization of cell walls, liberation of intracellular molecules, diffusivity, and selective extraction of bioactive compounds [52]. Optimization of ohmic heating, through an in-depth study of the effects of key independent variables on the response variables (extraction yield, total/individual phenolics, and bioactivity) potentially provides a value-added processing device for sustainable extraction of olive leaves.

## 4. Conclusions

Ohmic heating is considered a competitively viable approach for eco-sustainable extraction of biomolecules from food by-products. Using an inside-out heat generation, it overcomes the downsides inherent in conventional heating methods. The evidence presented in this study concludes that the ohmic technique is highly effective, compared to the conventional heating and control (solvent) methods, in increased extraction yield, total phenolics content, and antioxidant activity of olive leaves ($p < 0.001$). Of particular interest is that similar trends were seen in the results of TPC and antioxidant activity, irrespective of solvent ratio. Additional work will be needed to further understand the effects of different processing/operating parameters through an extensive comparative study between ohmic and competing/emerging methods. The ohmic technique, following optimization process may be employed as a competing benchmark for optimum extraction of olive leaves, enabling maximum/selective recovery of endogenous bio-phenols that can be used as high-value ingredients for bio-functional and nutraceutical applications.

Moreover, through a zero waste/sustainable process, it may address the challenges in current system for exploitation of olive leave residues which may highly assist in the improvement of the return of investment for agricultural/food producers.

**Author Contributions:** Conceptualization, F.S.M., J.A.T. and C.M.R.R.; Literature Reviewing, F.S.M., J.A.T. and C.M.R.R.; Methodology, F.S.M., J.A.T. and C.M.R.R.; Analysis and Interpretation of Results, F.S.M., J.A.T. and C.M.R.R.; Writing—original Draft: F.S.M., J.A.T. and C.M.R.R.; Writing—review and editing: F.S.M., J.A.T. and C.M.R.R.; Supervision, J.A.T. and C.M.R.R. All authors have read and agreed to the published version of the manuscript.

**Funding:** This work was supported by the Portuguese Foundation for Science and Technology (FCT), under the scope of the strategic funding of UID/BIO/04469/2020 unit.

**Institutional Review Board Statement:** Not applicable.

**Informed Consent Statement:** Not applicable.

**Data Availability Statement:** The data supporting the findings of this study are available within the article.

**Acknowledgments:** The authors wish to thank Eulogio Castro, Center for Advanced Studies in Energy and Environment, University of Jaén, Campus of Las Lagunillas, Jaén, Spain, for supplying the olive mill leaves.

**Conflicts of Interest:** The authors declare no conflict of interest.

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
