# Peer review of "Effect of Ohmic Heating on the Extraction Yield, Polyphenol Content and Antioxidant Activity of Olive Mill Leaves"

_cleantechnol, doi:10.3390/cleantechnol4020031_

Round 1

Reviewer 1 Report

THE MANUSCRIPT IS VERY INTRESTING, BUT IT NEEDS MINOR REVISIONS TO ACCEPT FOR PUBLICATION AS ATTACHED MY COMMENTS.

Author Response

Please see the attachment (thank you)

Reviewer 2 Report

 Effect of Ohmic Heating on the Extraction Efficiency and Anti-2 oxidant Activity of Polyphenols from Olive Mill Leaves

The work compares conventional and non-thermal extraction of polyphenols from olive leaves with ohmic heating assisted extraction. The main relevance is the application of ohmic heating for the polyphenol extraction, which has very few previous reports. The work is well written, clearly organized and presents the background that supports the proposal. Material and methods are well described, and the results are presented and discussed. The conclusion generalizes the work and proposes future investigations. I only have minor comments.

Since the stirrer was used for all the proposed methods, it has been considered as a standard procedure. However, it is well known that the mechanical input could affect the extraction process. Since it was not investigated in the current work, I suggest the authors to at least briefly discuss how it could possibly affect the extraction based on previous works, whether it could have some synergistic effect with ohmic heating. I hereby list a few references from a quick search that may provide support, as a suggestion, but the authors may feel free to use them to discuss the influence of mechanical treatment, or search for others:

Polyphenol Extraction by Di_erent Techniques for Valorisation of Non-Compliant Portuguese Sweet Cherries towards a Novel Antioxidant Extract - doi:10.3390/su12145556

Green Extraction Methods for Extraction of Polyphenolic Compounds from Blueberry Pomace - doi:10.3390/foods9111521

Minor comments

Table 1/2 – remove the horizontal lines between the components, keep only below the header

Conclusion: The conclusion could be shortened to the most relevant issues, with other comments being moved to the discussion section (for instance, lines 522-527).
